# Frontal cortex selects representations of the talker's mouth to aid in speech perception

**Muge Ozker[1]\*, Daniel Yoshor[1,2], Michael S Beauchamp[1]\***

[1]Department of Neurosurgery, Baylor College of Medicine, Texas, United States; [2]Michael E. DeBakey Veterans Affairs Medical Center, Texas, United States

**Abstract** Human faces contain multiple sources of information. During speech perception, visual information from the talker's mouth is integrated with auditory information from the talker's voice. By directly recording neural responses from small populations of neurons in patients implanted with subdural electrodes, we found enhanced visual cortex responses to speech when auditory speech was absent (rendering visual speech especially relevant). Receptive field mapping demonstrated that this enhancement was specific to regions of the visual cortex with retinotopic representations of the mouth of the talker. Connectivity between frontal cortex and other brain regions was measured with trial-by-trial power correlations. Strong connectivity was observed between frontal cortex and mouth regions of visual cortex; connectivity was weaker between frontal cortex and non-mouth regions of visual cortex or auditory cortex. These results suggest that top-down selection of visual information from the talker's mouth by frontal cortex plays an important role in audiovisual speech perception.

DOI: https://doi.org/10.7554/eLife.30387.001

**\*For correspondence:**
mozker@gmail.com (MO);
michael.beauchamp@bcm.edu
(MSB)

**Competing interests:** The authors declare that no competing interests exist.

## Introduction

Speech perception is multisensory: humans combine visual information from the talker's face with auditory information from the talker's voice to aid in perception. The contribution of visual information to speech perception is influenced by two factors. First, if the auditory information is noisy or absent, visual speech is more important than if the auditory speech is clear. Current models of speech perception assume that top-down processes serve to incorporate this factor into multisensory speech perception. For instance, visual cortex shows enhanced responses to audiovisual speech containing a noisy or entirely absent auditory component (*Schepers et al., 2015*) raising an obvious question: since visual cortex presumably cannot assess the quality of auditory speech on its own, what is the origin of the top-down modulation that enhances visual speech processing ? Neuroimaging studies have shown that speech reading (perception of visual-only speech) leads to strong responses in frontal regions including the inferior frontal gyrus, premotor cortex, frontal eye fields and dorsolateral prefrontal cortex (*Callan et al., 2014*; *Calvert and Campbell, 2003*; *Hall et al., 2005*; *Lee and Noppeney, 2011*; *Okada and Hickok, 2009*). We predicted that these frontal regions serve as the source of a control signal, which enhances activity in visual cortex when auditory speech is noisy or absent.

Second, visual information about the content of speech is not distributed equally throughout the visual field. Some regions of the talker's face are more informative about speech content, with the mouth of the talker carrying the most information (*Vatikiotis-Bateson et al., 1998*; *Yi et al., 2013*). If frontal cortex enhances visual responses during audiovisual speech perception, one should expect this enhancement to occur preferentially in regions of visual cortex which represent the mouth of the talker's face.

# Results

Responses to three types of speech—audiovisual (*AV*), visual-only (*Vis*) and auditory-only (*Aud*) (*Figure 1A*) —were measured in 73 electrodes implanted over visual cortex (*Figure 1B*). Neural activity was assessed using the broadband response amplitude (*Crone et al., 2001*; *Canolty et al., 2007*; *Flinker et al., 2015*; *Mesgarani et al., 2014*). The response to *AV* and *Vis* was identical early in the trial, rising quickly after stimulus onset and peaking at about 160% of baseline (*Figure 1C*); there was no response to *Aud* speech. The talker's mouth began moving at 200 ms after stimulus onset, with the onset of auditory speech in the *AV* condition occurring at 283 ms. Shortly thereafter (at 400 ms) the responses to *AV* and *Vis* diverged, with significantly greater responses to *Vis* than *AV*. Examining individual electrodes revealed large variability in the amount of *Vis* enhancement, ranging from −16 to 78% (*Figure 1D*).

Enhanced visual cortex responses to *Vis* speech may reflect the increased importance of visual speech for comprehension when auditory speech is not available. However, visual speech information is not evenly distributed across the face: the mouth of the talker contains most of the information about speech content. We predicted that mouth representations in visual cortex should show greater enhancement than non-mouth representations. To test this prediction, we measured the receptive field of the neurons underlying each electrode by presenting small checkerboards at different visual field locations and determining the location with the maximum evoked response (*Figure 2A*). As expected, electrodes located near the occipital pole had receptive fields near the center of the visual field, while electrodes located more anteriorly along the calcarine sulcus had receptive fields in the visual periphery (*Figure 2B*). In the speech stimulus, the mouth subtended approximately 5 degrees of visual angle. Electrodes with receptive fields of less than five degrees eccentricity were assumed to carry information about the mouth region of the talker's face and were classified as 'mouth electrodes' (*N* = 49). Electrodes with more peripheral receptive fields were classified as 'non-mouth electrodes' (*N* = 24). We compared the response in each group of electrodes using a linear mixed-effects (LME) model with the response amplitude as the dependent measure;

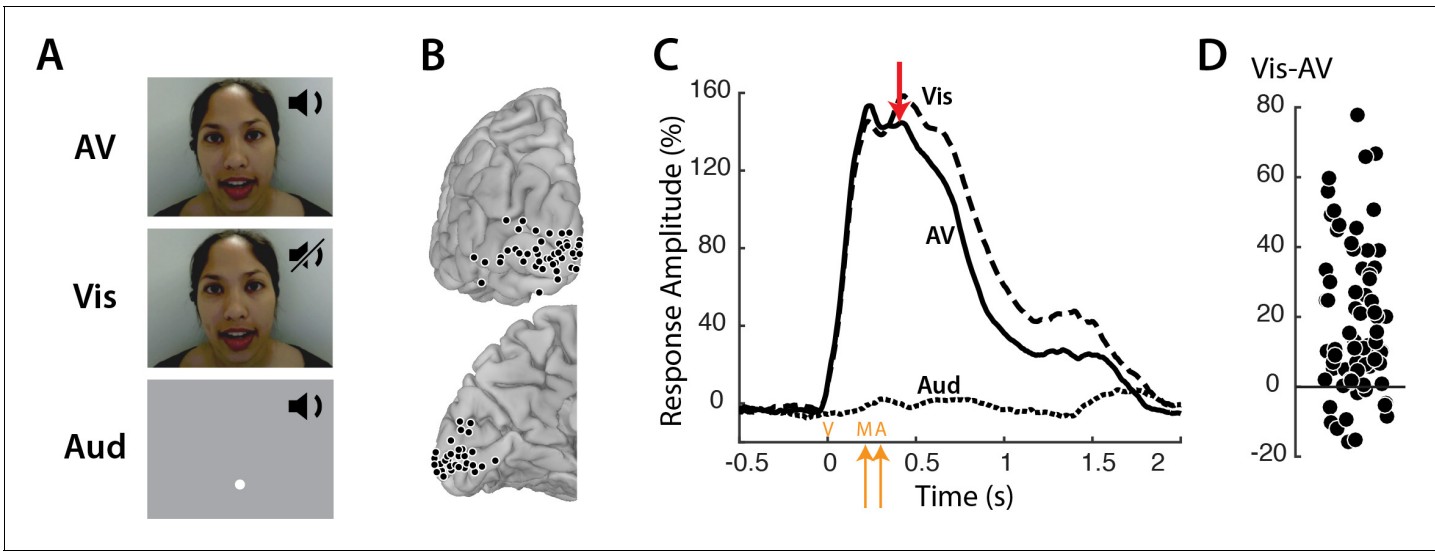

**Figure 1.** Visual cortex responses to speech. (A) The speech stimuli consisted of audiovisual recordings of a female talker speaking words (*AV*) edited so that only the visual portion of the recording was presented (*Vis*) or only the auditory portion of the recording was presented (*Aud*). Subjects were instructed to fixate the talker's mouth (*AV* and *Vis* conditions) or a fixation point presented at the same screen location as the talker's mouth (*Aud* condition). (B) In eight subjects, a total of 73 responsive occipital electrodes were identified. Electrode locations (black circles) are shown on a posterior view (top) and a medial view (bottom) of the left hemisphere of a template brain. (C) Broadband responses (70–150 Hz) to *AV* (solid line), *Vis* (dashed line) and *Aud* (dotted line) speech averaged across electrodes. The red arrow indicates the first time point (400 ms) at which the response to *Vis* speech was significantly greater than the response to *AV* speech. On the x-axis, time zero corresponds to the onset of the video (V) followed by the onset of the talker's mouth movements (M) at 200 ms and onset of auditory speech (A) at 283 ms (orange arrows). (D) The broadband response enhancement (*Vis* – *AV*) measured from 200 to 1500 ms. One symbol per electrode (symbols jittered along x-axis for improved visibility).

DOI: https://doi.org/10.7554/eLife.30387.002

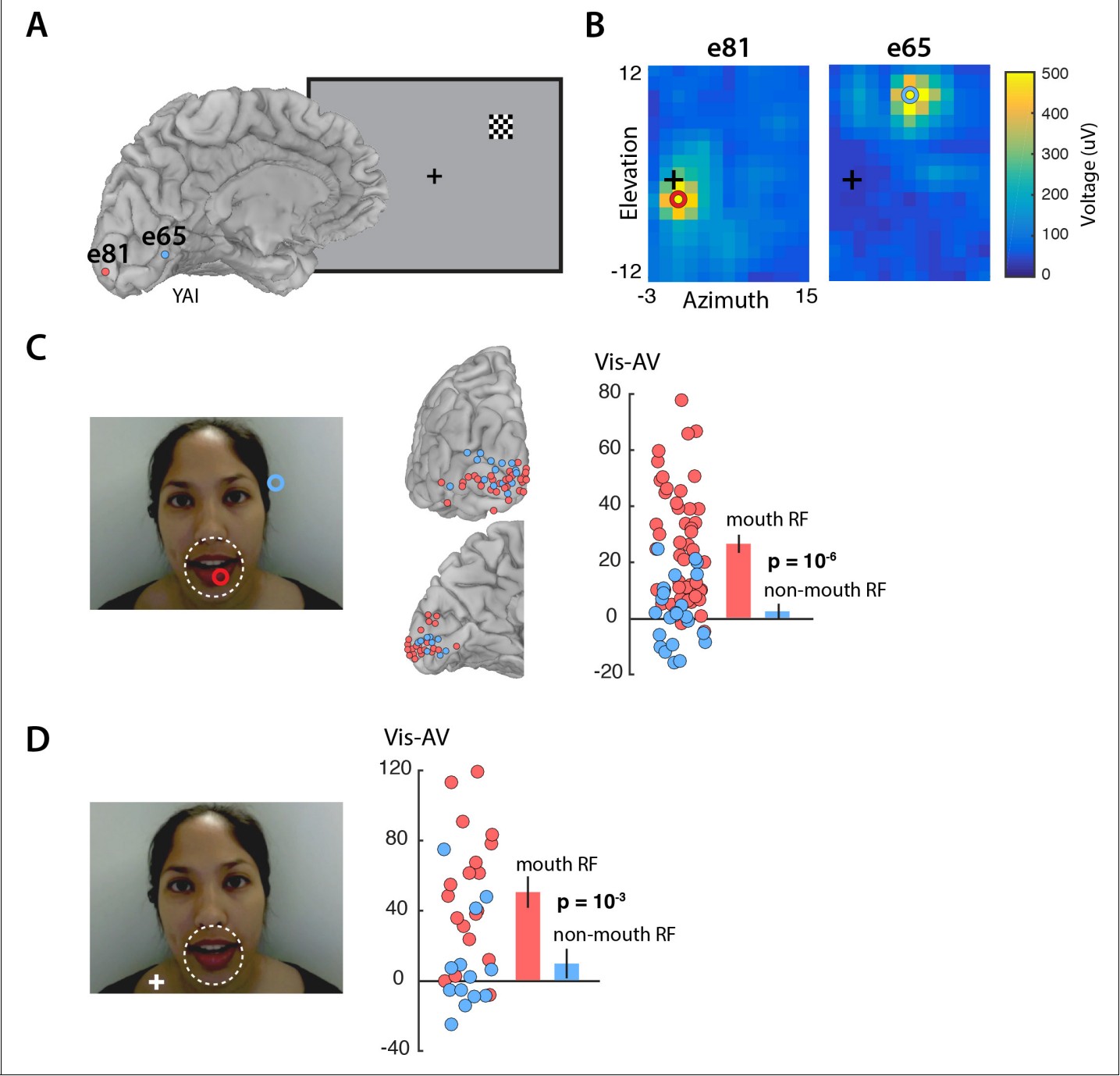

**Figure 2.** Retinotopic organization of speech responses in the visual cortex. (**A**) Medial view of a cortical surface model of the left hemisphere brain of a single subject (anonymized subject ID YAI). Posterior electrode e81 (red circle) was located superior to the calcarine sulcus on the occipital pole (red circle) while anterior electrode e65 (blue circle) was located inferior to the calcarine on the medial wall of the hemisphere. The receptive field mapping stimulus consisted of a small checkerboard presented at random screen locations while subjects performed a letter detection task at fixation (not shown). (**B**) The responses evoked by the mapping stimulus in electrodes e81 (left panel) and e65 (right panel). Color scales corresponds to the amplitude of the visual evoked response at each location in the visual field, with the crosshairs showing the center of the visual field and the red and blue circles showing the center of a two-dimensional Gaussian fitted to the response. Electrode e81 had a central receptive field (eccentricity at RF center of 2.5°) while electrode e65 had a peripheral receptive field (eccentricity 10.9°). (**C**) The receptive field location for the two sample electrodes, e81 (red circle) and e65 (blue circle) are shown on the speech stimulus. Subjects were instructed to fixate the talker's mouth and electrodes were classified as representing the mouth region of the talker's face (red circles; less than 5° from the center of the mouth, white dashed line) or as non-mouth (blue circles;>5°). Electrode locations of mouth (red) and non-mouth (blue) electrodes shown on posterior and medial brain views. Response enhancement
*Figure 2 continued on next page*

*Figure 2 continued*

(*Vis-AV*) for each individual electrode; inset bar graph shows mean values (±standard error). (**D**) In a control speech experiment, a white fixation crosshair was presented on the talker's shoulder, moving the mouth of the talker's face to the visual periphery. Electrodes were classified as representing the mouth of the talker (red circles; less than 5° from the the center of the mouth, white dashed line) or non-mouth (blue circles;>5°). Mouth electrodes were in the periphery of the visual field (mean eccentricity of 9.5°). Response enhancement (*Vis-AV*) for each individual electrode; inset bar graph shows mean values (±standard error).

DOI: https://doi.org/10.7554/eLife.30387.003

the electrode type (mouth or non-mouth) and stimulus condition (*AV*, *Vis* or *Aud*) as fixed factors; electrode as a random factor; and the response in mouth electrodes to *AV* speech as the baseline (see *Table 1* for all LME results).

The model found a significant main effect of stimulus condition, driven by greater responses to *Vis* (95% *vs.* 77%, *Vis vs. AV*; p=$10^{-6}$) and weaker responses to *Aud* (12% *vs.* 77%, *Aud* vs. *AV*; p=$10^{-16}$). Critically, there was also a significant interaction between stimulus condition and electrode type. Mouth electrodes showed a large difference between the responses to *Vis* and *AV* speech while non-mouth electrodes showed almost no difference (26% *vs.* 2%, mouth *vs.* non-mouth *Vis – AV*; p=0.01; *Figure 2C*).

An alternative interpretation of this results is raised by the task design, in which subjects were instructed to fixate the mouth of the talker. Therefore, the enhancement could be specific to electrodes with receptive fields near the center of gaze (central *vs.* peripheral) rather than to the mouth of the talker's face (mouth *vs.* non-mouth).

To distinguish these two possibilities, we performed a control experiment in which the task instructions were to fixate the shoulder of the talker, rather than the mouth, aided by a fixation crosshair superimposed on the talker's shoulder (*Figure 2D*). With this manipulation, the center of the mouth was located at 10 degrees eccentricity. Electrodes with receptive field centers near the mouth were classified as mouth electrodes (*N* = 19); otherwise, they were classified as non-mouth electrodes (*N* = 13). This dissociated visual field location from mouth location: mouth electrodes had receptive fields located peripherally (mean eccentricity = 9.5 degrees). If the enhancement during *Vis* speech was restricted to the center of gaze, we would predict no enhancement for mouth electrodes in this control experiment due to their peripheral receptive fields. Contrary to this prediction, the LME showed a significant interaction between electrode type and stimulus condition (*Table 2*). Mouth electrodes (all with peripheral receptive fields) showed a large difference between the responses to *Vis* and *AV* speech while non-mouth electrodes showed little difference (50% *vs.* 9%, mouth *vs.* non-mouth *Vis – AV*; p=$10^{-3}$).

Both experiments demonstrated a large enhancement for *Vis* compared with *AV* speech in visual cortex electrodes representing the mouth of the talker. However, the visual stimulus in the *Vis* and *AV* conditions was identical. Therefore, a control signal sensitive to the absence of auditory speech must trigger the enhanced visual responses. We investigated responses in frontal cortex as a possible source of top-down control signals (*Corbetta and Shulman, 2002*; *Gunduz et al., 2011*; *Kastner and Ungerleider, 2000*; *Miller and Buschman, 2013*).

*Figure 3A* shows the responses during *Vis* speech for a frontal electrode located at the intersection of the precentral gyrus and middle frontal gyrus (Talairach co-ordinates: x = −56, y = −8,

**Table 1.** LME for Amplitude in Visual Cortex

| Fixed effects: | Estimate | Std. error | DF | t-value | p-value |
|---|---|---|---|---|---|
| Baseline | 98 | 8.7 | 90.5 | 11.2 | $10^{-16}$ |
| A Speech | −83 | 5.9 | 134.5 | −14.2 | $10^{-16}$ |
| A Speech x Peripheral RF | 47 | 9.7 | 133.5 | 4.9 | $10^{-6}$ |
| V Speech | 26 | 5.4 | 132.9 | 4.8 | $10^{-6}$ |
| Peripheral RF | −65 | 15.2 | 90.5 | −4.3 | $10^{-5}$ |
| V Speech x Peripheral RF | −24 | 9.4 | 132.9 | −2.6 | 0.01 |

DOI: https://doi.org/10.7554/eLife.30387.004

**Table 2.** LME for Amplitude in Visual Cortex (Control Experiment)

| Fixed effects: | Estimate | Std. error | DF | t-value | p-value |
|---|---|---|---|---|---|
| Baseline | 357 | 35 | 32.7 | 10.1 | $10^{-11}$ |
| V Speech | 50 | 7 | 32 | 6.7 | $10^{-7}$ |
| Non-mouth RF | −287 | 55 | 32.7 | -5 | $10^{-5}$ |
| V Speech x Non-mouth RF | −41 | 12 | 32 | −3.5 | $10^{-3}$ |

DOI: https://doi.org/10.7554/eLife.30387.005

z = 33) and a visual electrode located on the occipital pole with a receptive field located in the mouth region of the talker's face (1.8° eccentricity). Functional connectivity was measured with a Spearman rank correlation of the trial-by-trial response power within each pair. This pair of electrodes showed strong correlation (ρ = 0.57, p=$10^{-6}$): on trials in which the frontal electrode responded strongly, the visual electrode did as well.

We predicted that connectivity should be greater for mouth electrodes than non-mouth electrodes. In each subject, we selected the single frontal electrode with the strongest response to *AV* speech (*Figure 3B*; average Talairach co-ordinates of the frontal electrode: x = 49, y = −2, z = 42) and measured the connectivity between the frontal electrode and all visual electrodes in that subject (*Figure 3B*).

An LME model (*Table 3*) was fit with the Spearman rank correlation (ρ) between each frontal-visual electrode pair as the dependent measure; the visual electrode type (mouth *vs.* non-mouth) and stimulus condition (*AV*, *Vis* or *Aud*) as fixed factors; electrode as a random factor; and connectivity of mouth electrodes during *AV* speech as the baseline. The largest effect in the model was a main effect of electrode type, driven by strong frontal-visual connectivity in mouth electrodes and

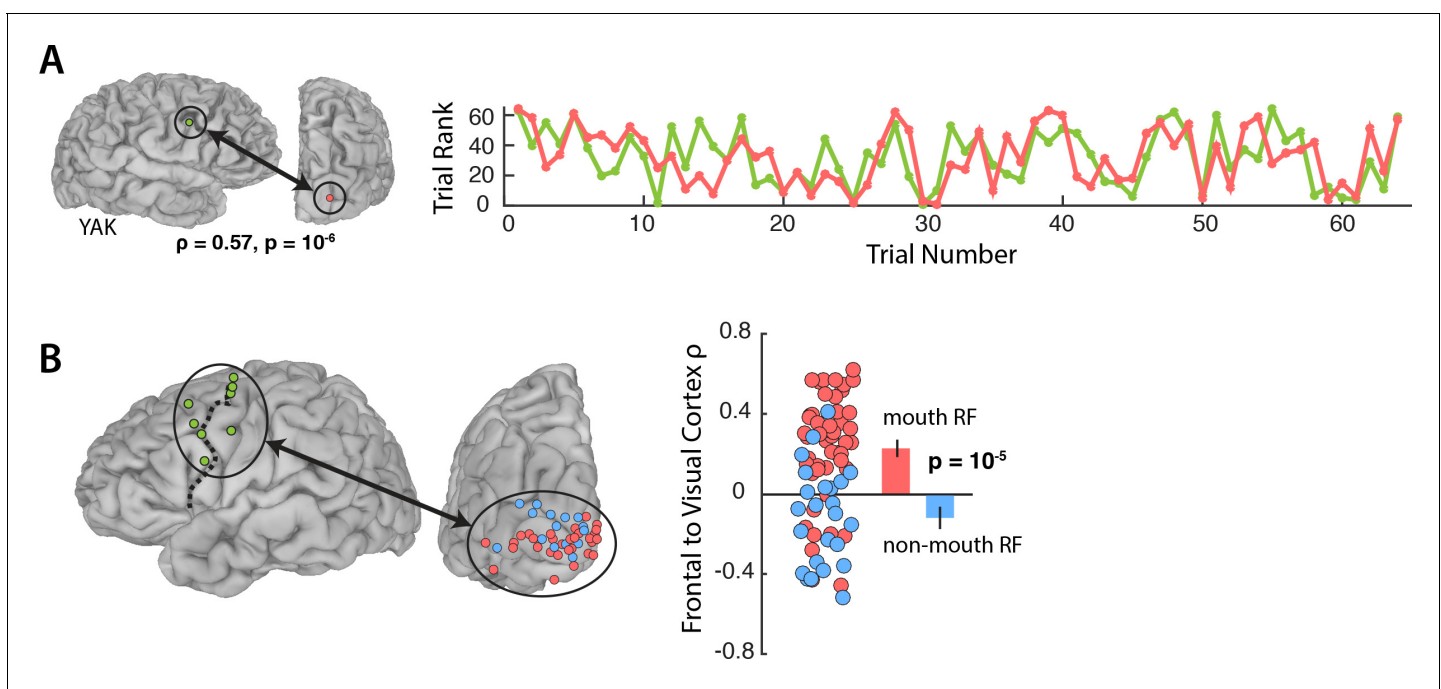

**Figure 3.** Functional connectivity with the frontal cortex. (**A**) Average broadband responses for each *Vis* speech trial (200–1500 ms, 70–150 Hz) measured simultaneously in a single frontal (green) and a single visual cortex (red) electrode. Trials were ranked based on their response amplitudes (y axis) and shown with respect to their presentation orders (x axis). The Spearman rank correlation between the amplitude time series of the two electrodes was high (ρ = 0.57). (**B**) Left hemisphere of a template brain showing all frontal (green circles), visual cortex mouth (red) and non-mouth (blue) electrodes. Frontal electrodes were located near the precentral sulcus (dashed line). Plot shows Spearman rank correlation between each frontal-visual electrode pair; inset bar graph shows mean values (±standard error).

DOI: https://doi.org/10.7554/eLife.30387.006

**Table 3.** LME for Frontal-Visual Cortex Connectivity

| Fixed effects: | Estimate | Std. error | DF | t-value | p-value |
|---|---|---|---|---|---|
| Baseline | 0.23 | 0.03 | 179.1 | 7.4 | $10^{-12}$ |
| Peripheral RF | −0.24 | 0.05 | 179.1 | −4.5 | $10^{-5}$ |
| A Speech | −0.1 | 0.04 | 141.1 | −2.5 | 0.01 |
| A Speech x Peripheral RF | 0.16 | 0.07 | 136.5 | 2.4 | 0.02 |
| V Speech x Peripheral RF | −0.1 | 0.06 | 134 | −1.6 | 0.1 |
| V Speech | 0.0003 | 0.04 | 134 | −0.007 | 1 |

DOI: https://doi.org/10.7554/eLife.30387.007

weak connectivity in non-mouth electrodes (0.2 *vs.* −0.02; p=$10^{-5}$). There was significantly weaker frontal-visual connectivity during the *Aud* condition in which no visual speech was presented (p=0.01).

Our results suggest a model in which frontal cortex modulates mouth regions of visual cortex in a task-specific fashion. Since top-down modulation is an active process that occurs when visual speech is most relevant, this model predicts that frontal cortex should be more active during the perception of *Vis* speech. To test this prediction, we compared the response to the different speech conditions in frontal cortex (*Figure 4A*). Frontal electrodes responded strongly to all three speech conditions, with a peak amplitude of 109% at 470 ms after stimulus onset. Following the onset of auditory speech (283 ms after stimulus onset), the responses diverged, with a larger response to *Vis* than *AV* or *Aud* speech (53% *vs.* 33%, *Vis vs. AV*; p=0.001; Table 6). Greater responses to *Vis* speech were consistent across frontal electrodes (*Figure 4B*).

Both frontal and visual cortex showed enhanced responses to *Vis* speech. If frontal cortex was the source of top-down modulation that resulted in visual cortex enhancements, one might expect frontal response differences to *precede* visual cortex response differences. To test this idea, we examined the time courses of the response to speech in a sample frontal-visual electrode pair (*Figure 4C*). The increased response to *Vis* speech occurred *earlier* in the frontal electrode, beginning at 247 ms after auditory onset versus 457 ms for the visual electrode. To test whether the earlier divergence between *Vis* and *AV* speech in frontal compared with visual cortex was consistent across electrodes, for each electrode we calculated the first time point at which the *Vis* response was significantly larger than the *AV* response (*Figure 4D*). 27 out of 29 visual electrodes showed *Vis* response enhancement after their corresponding frontal electrode, with an average latency difference of 230 ms (frontal *vs.* visual: 257 ± 150 ms *vs.* 487 ± 110 ms *vs.* with respect to auditory speech onset, $t_{34} = 4$, p=$10^{-4}$).

While our primary focus was on frontal modulation of visual cortex, auditory cortex was analyzed for comparison. 44 electrodes located on the superior temporal gyrus responded to *AV* speech (*Figure 5A*). As expected, these electrodes showed little response to *Vis* speech but a strong response to *Aud* speech, with a maximal response amplitude of 148% at 67 ms after auditory speech onset. While visual cortex showed a greater response to *Vis* speech than to *AV* speech, there was no significant difference in the response of auditory cortex to *Aud* and *AV* speech, as confirmed by the LME model (p=0.8, *Table 4*). Next, we analyzed the connectivity between frontal cortex and auditory electrodes. The LME revealed a significant effect of stimulus, driven by weaker connectivity in the *Vis* condition in which no auditory speech was presented (p=0.02; *Table 5*). Overall, the connectivity between frontal cortex and auditory cortex was weaker than the connectivity between frontal cortex and visual cortex (0.04 for frontal-auditory *vs.* 0.23 for frontal-visual mouth electrodes; p=0.001, unpaired t-test).

In visual cortex, a subset of electrodes (those representing the mouth) showed a greater response in the *Vis-AV* contrast and greater connectivity with frontal cortex. To determine if the same was true in auditory cortex, we selected the STG electrodes with the strongest response in the *Aud-AV* contrast (*Figure 5B*). Unlike in visual cortex, in which there was anatomical localization of mouth-representing electrodes on the occipital pole, auditory electrodes with the strongest response in the *Aud-AV* contrast did not show clear anatomical clustering (*Figure 5C*). Electrodes with the electrodes with the strongest response in the *Aud-AV* contrast had equivalent connectivity with frontal

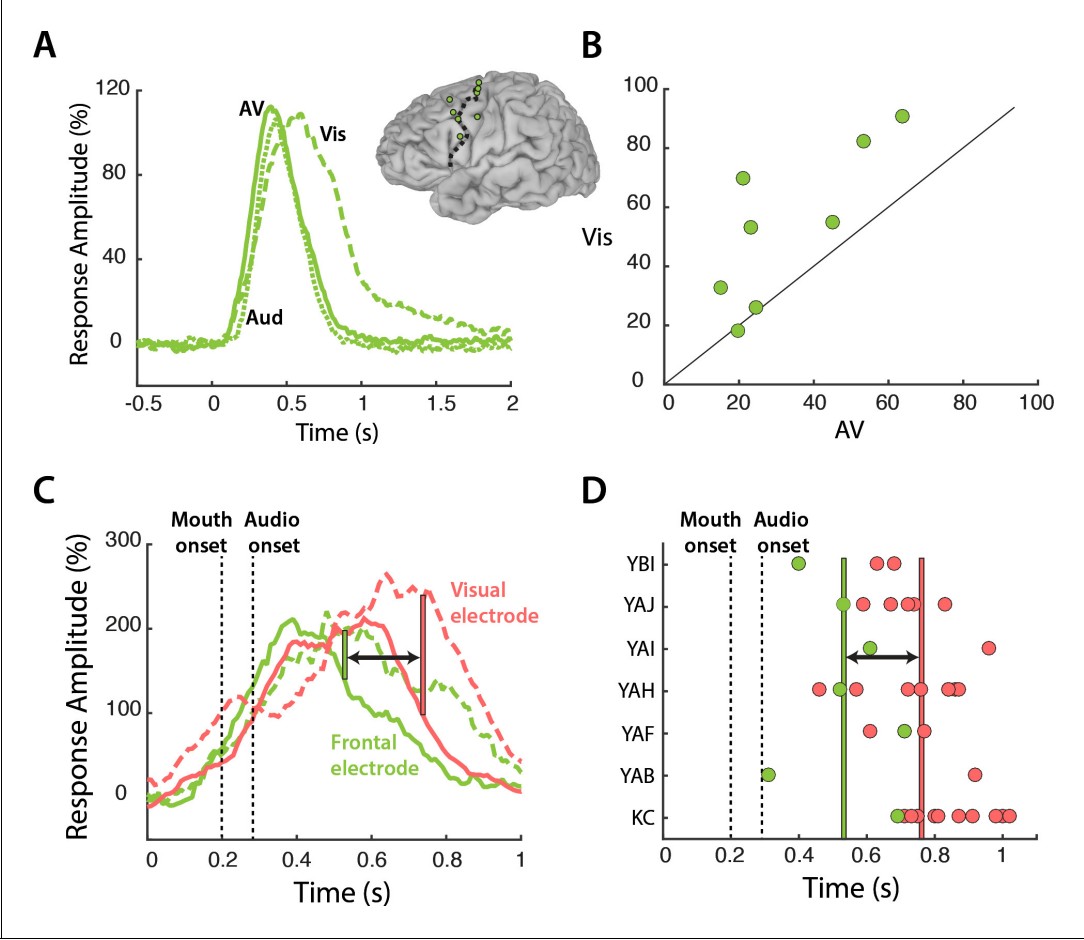

**Figure 4.** Frontal cortex responses. (**A**) Broadband responses to AV (solid line), Vis (dashed line) and Aud (dotted line) speech averaged across frontal electrodes (locations shown on inset brain). (**B**) Scatter plot showing average responses (from 200 to 1500 ms) to Vis and AV speech for all frontal electrodes (green circles). Black diagonal line indicates the line of equality. (**C**) Broadband responses to AV (solid line) and Vis (dashed line) speech in a single frontal (green trace) and visual cortex (red trace) electrode. Dashed black lines depicts the onset of mouth movements and the onset of auditory speech (note expanded scale on x-axis compared with A). Vertical green line indicates the first time point at which the frontal response to Vis speech was enhanced relative to AV speech. Red line indicates the same for visual cortex. Black arrow highlights latency difference. (**D**) Enhancement latency for all frontal (green circles) and visual (red circles) electrodes. Letter codes along y-axis show different subjects. Vertical green and red lines indicate the average enhancement latency across all frontal and visual electrodes respectively.

DOI: https://doi.org/10.7554/eLife.30387.008

cortex as other STG electrodes (ρ = −0.04 *vs. roh* = 0.06, unpaired t-test = 0.9, p=0.3). This was the opposite of the pattern observed in visual cortex. In order to ensure that signal amplitude was not the main determinant of connectivity, we selected the STG electrodes with the highest signal-to-noise ratio in the AV condition. These electrodes had equivalent connectivity with frontal cortex as other STG electrodes (ρ = −0.002 *vs.* ρ = 0.05, unpaired t-test = 0.4, p=0.7).

## Discussion

Taken together, these results support a model in which control regions of lateral frontal cortex located near the precentral sulcus selectively modulate visual cortex, enhancing activity with both spatial selectivity—only mouth regions of the face are enhanced —and context selectivity—enhancement is greater when visual speech is more important due to the absence of auditory speech.

Our results link two distinct strands of research. First, fMRI studies of speech perception frequently observe activity in frontal cortex, especially during perception of a visual-only speech, a task sometimes referred to as speech reading (*Callan et al., 2014*; *Calvert and Campbell, 2003*;

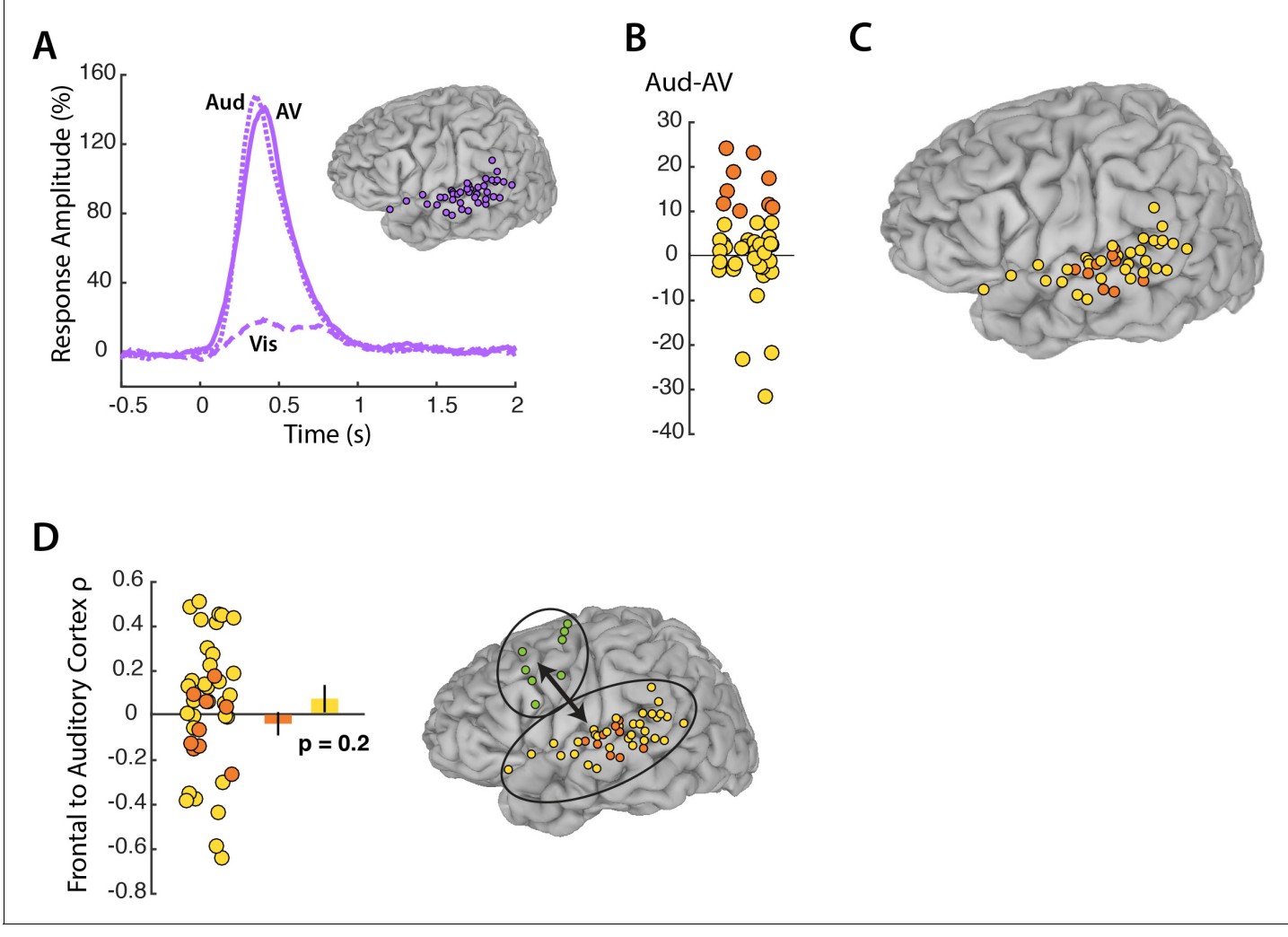

**Figure 5.** Auditory cortex responses and connectivity. (A) Broadband responses to *AV* (solid line), *Vis* (dashed line) and *Aud* (dotted line) speech across auditory electrodes located on the STG (purple circles on inset brain). (B) The broadband response enhancement (*Aud − AV*) with one symbol per electrode (symbols jittered along x-axis for improved visibility). Electrodes were divided into two groups, those showing a large (>=10%) enhancement (orange circles) and those showing little or no enhancement (yellow circles). (C) Anatomical distribution of STG electrodes by *Aud − AV* amplitude. (D) Frontal connectivity of STG electrodes by *Aud − AV* amplitude.

DOI: https://doi.org/10.7554/eLife.30387.009

*Hall et al., 2005*; *Lee and Noppeney, 2011*; *Okada and Hickok, 2009*). However, the precise role of this frontal activity has been difficult to determine given the relatively slow time resolution of fMRI.

Second, it is well known that frontal regions in an around the frontal eye fields in the precentral sulcus modulate visual cortex activity during tasks that require voluntary control of spatial or featural attention (*Corbetta and Shulman, 2002*; *Gregoriou et al., 2012*; *Gunduz et al., 2011*; *Kastner and*

**Table 4.** LME for Amplitude in Auditory Cortex

| Fixed effects: | Estimate | Std. error | DF | t-value | p-value |
|---|---|---|---|---|---|
| Baseline | 41 | 4.9 | 88.3 | 8.4 | $10^{-12}$ |
| V Speech | −32 | 5 | 83.2 | −6.4 | $10^{-8}$ |
| A Speech | 1 | 5.1 | 84.1 | 0.2 | 0.8 |

DOI: https://doi.org/10.7554/eLife.30387.010

**Table 5.** LME for Frontal-Auditory Cortex Connectivity

| Fixed effects: | Estimate | Std. error | DF | t-value | p-value |
|---|---|---|---|---|---|
| Baseline | 0.04 | 0.04 | 83.6 | 1.1 | 0.3 |
| V Speech | −0.1 | 0.04 | 83.6 | −2.3 | 0.02 |
| A Speech | 0.02 | 0.04 | 84.4 | 0.4 | 0.7 |

DOI: https://doi.org/10.7554/eLife.30387.011

*Ungerleider, 2000*; *Miller and Buschman, 2013*; *Popov et al., 2017*). However, it has not been clear how these attentional networks function during other important cognitive tasks, such as speech perception.

We suggest that frontal-visual attentional control circuits are automatically engaged during speech perception in the service of increasing perceptual accuracy for the processing of this very important class of stimuli. This allows for precise, time-varying control: as the quality of auditory information fluctuates as auditory noise in the environment increases or decreases, frontal cortex can up or down-regulate activity in visual cortex accordingly. It also allows for precise spatial control: as the mouth of the talker contains the most speech information, frontal cortex can selectively enhance visual cortex activity that is relevant for speech perception by enhancing activity in subregions of visual cortex that represent the talker's mouth. A possible anatomical linkage supporting this processing is the inferior fronto-occipital fasciculus connecting frontal and occipital regions, found in human but not non-human primates (*Catani and Mesulam, 2008*). Most models of speech perception focus on auditory cortex inputs into parietal and frontal cortex (*Hickok and Poeppel, 2004*; *Rauschecker and Scott, 2009*). Our findings suggest that visual cortex should also be considered an important component of the speech perception network, as it is selectively and rapidly modulated during audiovisual speech perception.

The analysis of auditory cortex responses provides an illuminating contrast with the visual cortex results. While removing auditory speech increased visual cortex response amplitudes and frontal-visual connectivity, removing visual speech did not change auditory cortex response amplitudes or connectivity. A simple explanation for this is that auditory speech is easily intelligible without visual speech, so that no attentional modulation is required. In contrast, perceiving visual-only speech requires speechreading, a difficult task that demands attentional engagement. An interesting test of this idea would be to present auditory speech with added noise. Under these circumstances, attentional engagement would be expected in order to enhance the intelligibility of the noisy auditory speech, with a neural manifestation of increased response amplitudes in auditory cortex and increased connectivity with frontal cortex. The interaction of stimulus and task also provides an explanation for the frontal activations in response to the speech stimuli. The human frontal eye fields show rapid sensory-driven responses, with latencies as short as 24 ms to auditory stimuli and 45 ms to visual stimuli (*Kirchner et al., 2009*). Following initial sensory activation, task demands modulate frontal function, with demanding tasks such as processing visual-only or noisy auditory speech or resulting in enhanced activity (*Cheung et al., 2016*; *Cope et al., 2017*).

**Table 6.** LME for Amplitude in Frontal Cortex

| Fixed effects: | Estimate | Std. error | DF | t-value | p-value |
|---|---|---|---|---|---|
| Baseline | 33 | 6.7 | 12.1 | 5 | $10^{-4}$ |
| V Speech | 20 | 5.2 | 14.1 | 3.9 | $10^{-3}$ |
| A Speech | -7 | 5.8 | 14.4 | −1.1 | 0.3 |

DOI: https://doi.org/10.7554/eLife.30387.012

## Materials and methods

### Subject information

All experimental procedures were approved by the Institutional Review Board of Baylor College of Medicine. Eight human subjects provided written informed consent prior to participating in the research protocol. The subjects (5F, mean age 36, 6L hemisphere) suffered from refractory epilepsy and were implanted with subdural electrodes guided by clinical requirements. Following surgery, subjects were tested while resting comfortably in their hospital bed in the epilepsy monitoring unit.

### Experiment setup

Visual stimuli were presented on an LCD monitor (Viewsonic VP150, 1024 × 768 pixels) positioned at 57 cm distance from the subject, resulting in a display size of 30.5° x 22.9°.

### Receptive field mapping procedures

Mapping stimulus consisted of a square checkerboard pattern (3° x 3° size) briefly flashed (rate of 2 Hz and a duty cycle of 25%) in different positions on the display monitor to fill a grid over the region of interest in the visual field (63 positions, 7 × 9 grid). 12–30 trials for each position were recorded.

Subjects fixated at the center of the screen and performed a letter detection task to ensure that they were not fixating on the mapping stimulus. Different letters were randomly presented at the center of the screen (2° in size presented at a rate of 1–4 Hz) and subjects were required to press a mouse button whenever the letter 'X' appeared. The mean accuracy was 88 ± 14% with a false alarm rate of 8 ± 14% (mean across subjects ± SD; responses were not recorded for one subject).

### Speech experiment procedures

Four video clips of a female talker pronouncing the single syllable words 'drive', 'known', 'last' and 'meant' were presented under audiovisual (*AV*), visual (*Vis*) and auditory (*Aud*) conditions. Visual stimuli were presented using the same monitor used for receptive field mapping, with the face of the talker subtending approximately 13 degrees horizontally and 21 degrees vertically. Speech sounds were played through loudspeakers positioned next to the subject's bed. The average duration of the video clips was ~1500 ms (drive: 1670 ms, known: 1300 ms, last: 1500 ms, meant: 1400 ms). In *AV* and *Vis* trials, mouth movements started at ~200 ms after the video onset on average (drive: 200 ms, known: 233 ms, last: 200 ms, meant: 200 ms). In *AV* trials, auditory vocalizations started at ~283 ms (drive: 267 ms, known: 233 ms, last: 300 ms, meant: 333 ms). Vocalization duration was ~480 ms on average (drive: 500 ms, known: 400 ms, last: 530 ms, meant: 500 ms).

The three different conditions were randomly intermixed, separated by interstimulus intervals of 2.5 s. 32–64 repetitions for each condition was presented. Subjects were instructed to fixate either the mouth of the talker (during *Vis* and *AV* trials) or a white fixation dot presented at the same location as the mouth of the talker on a gray background (during *Aud* trials and the interstimulus intervals). To ensure attention to the stimuli, subjects were instructed to press a mouse button 20% of trials in which a catch stimulus was presented, consisting of the AV word 'PRESS'. The mean accuracy was 88 ± 18%, with a false alarm rate of 3 ± 6% (mean across subjects ± SD; for one subject, button presses were not recorded).

### Control speech experiment procedures

In a control experiment (one subject, 32 electrodes) identical procedures were used except that the subject fixated crosshairs placed on the shoulder of the talker (*Figure 2D*).

### Electrode localization and recording

Before surgery, T1-weighted structural magnetic resonance imaging scans were used to create cortical surface models with FreeSurfer, RRID:SCR_001847 (*Dale et al., 1999*; *Fischl et al., 1999*) and visualized using SUMA (*Argall et al., 2006*) within the Analysis of Functional Neuroimages package, RRID:SCR_005927 (*Cox, 1996*). Subjects underwent a whole-head CT after the electrode implantation surgery. The post-surgical CT scan and pre-surgical MR scan were aligned using and all electrode positions were marked manually on the structural MR images. Electrode positions were then projected to the nearest node on the cortical surface model using the AFNI program *SurfaceMetrics*.

Resulting electrode positions on the cortical surface model were confirmed by comparing them with the photographs taken during the implantation surgery.

A 128-channel Cerebus amplifier (Blackrock Microsystems, Salt Lake City, UT) was used to record from subdural electrodes (Ad-Tech Corporation, Racine, WI) that consisted of platinum alloy discs embedded in a flexible silicon sheet. Two types of electrodes were implanted, containing an exposed surface of either 2.3 mm or 0.5 mm; an initial analysis did not suggest any difference in the responses recorded from the two types of electrodes, so they were grouped together for further analysis. An inactive intracranial electrode implanted facing the skull was used as a reference for recording. Signals were amplified, filtered (low-pass: 500 Hz, fourth-order Butterworth filter; high-pass: 0.3 Hz, first-order Butterworth) and digitized at 2 kHz. Data files were converted from Blackrock format to MATLAB 8.5.0 (MathWorks Inc. Natick, MA) and the continuous data stream was epoched into trials. All analyses were conducted separately for each electrode.

## Receptive field mapping analysis: Evoked potentials

The voltage signal in each trial (consisting of the presentation of a single checkerboard at a single spatial location) was filtered using a Savitzky-Golay polynomial filter (''sgolayfilt'' function in Matlab) with polynomial order set to five and frame size set to 11. If the raw voltage exceeded a threshold of 3 standard deviations from the mean voltage, suggesting noise or amplifier saturation, the trial was discarded; <1 trial per electrode discarded on average. The filtered voltage response at each spatial location was averaged, first across trials and then across time-points (from 100 to 300 ms post-stimulus) resulting in a single value for response amplitude; these values were then plotted on a grid corresponding to the visual field (*Figure 1A*). A two-dimensional Gaussian function was fit to the responses to approximate the average receptive field of the neurons underlying the electrode. The center of the fitted Gaussian was used as the estimate of the RF center of the neurons underlying the electrode. A high correlation between the fitted Gaussian and the raw evoked potentials indicated an electrode with a high-amplitude, focal receptive field. A threshold of $r > 0.7$ was used to select only these electrodes for further consideration (*Yoshor et al., 2007*).

## Speech stimuli analysis: Broadband power

While for the RF mapping analysis, we used raw voltage as our measure of neural response, speech stimuli evoke a long-lasting response not measurable with evoked potentials. Therefore, our primary measure of neural activity was the broadband (non-synchronous) response in the high-gamma frequency band, ranging from 70 to 150 Hz. This response is thought to reflect action potentials in nearby neurons (*Jacques et al., 2016*; *Mukamel et al., 2005*; *Nir et al., 2007*; *Ray and Maunsell, 2011*). To calculate broadband power, the average signal across all electrodes was subtracted from each individual electrode's signal (common average referencing), line noise at 60, 120, 180 Hz was filtered and the data was transformed to time–frequency space using the multitaper method available in the FieldTrip toolbox (*Oostenveld et al., 2011*) with 3 Slepian tapers; frequency window from 10 to 200 Hz; frequency steps of 2 Hz; time steps of 10 ms; temporal smoothing of 200 ms; frequency smoothing of ±10 Hz.

The broadband response at each time point following stimulus onset was measured as the percent change from baseline, with the baseline calculated over all trials and all experimental conditions in a time window from −500 to −100 ms before stimulus onset. To reject outliers, if at any point following stimulus onset the response was greater than ten standard deviations from the mean calculated across the rest of the trials, the entire trial was discarded (average of 10 trials were discarded per electrode, range from 1 to 16).

## Visual cortex electrode selection

Across eight subjects, we recorded from 154 occipital lobe electrodes. These were winnowed using two criteria. First, a well-demarcated spatial receptive field (see *Receptive Field Mapping Analysis* section, above). Second, a significant ($q < 0.01$, false-discovery rate corrected) broadband response to *AV* speech; because only the response to *AV* trials was used to select electrodes, we could measure the response to the *Aud* and *Vis* conditions without bias. Out of 154 total occipital lobe electrodes, 73 electrodes met both criteria.

### Auditory cortex electrode selection

We recorded from 102 electrodes located on the superior temporal gyrus. 44 out of 102 electrodes showed a significant ($q < 0.01$, false-discovery rate corrected) broadband response to *AV* speech.

### Frontal cortex electrode selection

We recorded from 179 electrodes located in lateral frontal cortex, defined as the lateral convexity of the hemisphere anterior to the central sulcus. 44 out of 179 electrodes showed a significant ($q < 0.01$, false-discovery rate corrected) broadband response to *AV* speech. In each of the eight subjects, we selected the single frontal electrode that showed the largest broadband response to *AV* speech, measured as the signal-to-noise ratio ($\mu/\sigma$).

### Linear mixed effects modeling

We used the *lme4* package, RRID:SCR_015654 (*Bates et al., 2014*) for the R Project for Statistical Computing, RRID:SCR_001905 to perform linear mixed effect (LME) analyses. Complete details of each analysis shown in *Tables.* Similar to an ANOVA (but allowing for missing data), the LME estimated the effect of each factor in units of the dependent variable (equivalent to beta weights in a linear regression) relative to an arbitrary baseline condition (defined in our analysis as the response to *AV* speech) and a standard error.

### Functional connectivity analysis

The average high-gamma power in the 200–1500 milliseconds was calculated for each trial, corresponding to this time in which mouth movements occur in the speech stimuli. After calculating the average broadband (70–150 Hz) power for each trial, functional connectivity between the 73 frontal-visual cortex as well as 44 frontal-auditory electrode pairs was measured by calculating the trial-by-trial Spearman rank correlation across trials of the same speech condition (*AV*, *Vis* or *Aud*) (*Foster et al., 2015*; *Hipp et al., 2012*).

### Time point of enhancement onset analysis

The average response showed a long-lasting enhancement for visual speech (*Figure 1C*). In order to measure the time at which this occurred in individual electrodes, we compared the time course of the response to *Vis* and *AV* condition. The onset of enhancement was defined as the first time point in which there was a long-lasting (>=200 ms) significantly greater response (p<0.05 using a running t-test) to *Vis* compared with *AV* speech. Using these criteria, we were able to measure the enhancement onset time in 7 frontal and 29 visual electrodes (*Figure 4D*).

## Acknowledgements

This research was funded by Veterans Administration Clinical Science Research and Development Merit Award Number 1I01C × 000325–01A1, NIH R01NS065395 U01NS098976.

## Additional information

### Funding

| Funder | Grant reference number | Author |
| --- | --- | --- |
| Veterans Administration Clinical Science Research and Development | Merit Award Number 1I01CX000325 | Daniel Yoshor |
| National Institutes of Health | R01NS065395 | Michael S Beauchamp |
| National Institutes of Health | U01NS098976 | Michael S Beauchamp |

The funders had no role in study design, data collection and interpretation, or the decision to submit the work for publication.

## Author contributions

Muge Ozker, Conceptualization, Data curation, Software, Formal analysis, Validation, Investigation, Visualization, Methodology, Writing—original draft; Daniel Yoshor, Resources, Supervision, Funding acquisition, Writing—review and editing; Michael S Beauchamp, Conceptualization, Resources, Supervision, Funding acquisition, Validation, Investigation, Visualization, Methodology, Writing—original draft, Project administration, Writing—review and editing

## Author ORCIDs

Muge Ozker  http://orcid.org/0000-0001-7472-4528

Michael S Beauchamp  http://orcid.org/0000-0002-7599-9934

## Ethics

Human subjects: All experimental procedures were approved by the Institutional Review Board of Baylor College of Medicine. Eight human subjects provided written informed consent prior to participating in the research protocol. The experimenter who recorded the stimuli used in the experiments gave written authorization for her likeness to be used for illustrating the stimuli in Figures 1 and 2 of the manuscript.

## Decision letter and Author response

Decision letter https://doi.org/10.7554/eLife.30387.015

Author response https://doi.org/10.7554/eLife.30387.016

## Additional files

### Supplementary files

• Transparent reporting form

DOI: https://doi.org/10.7554/eLife.30387.013

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
