## [Decision Letter]

[Editors’ note: this article was originally rejected after discussions between the reviewers, but the authors were invited to resubmit after an appeal against the decision.]

Thank you for submitting your work entitled "Retinotopic Modulation of Visual Responses to Speech by Frontal Cortex Measured with Electrocorticography" for consideration by *eLife*. Your article has been reviewed by two peer reviewers, and the evaluation has been overseen by a Reviewing Editor and a Senior Editor. The following individuals involved in review of your submission have agreed to reveal their identity: Edward F Chang (Reviewer #2).

Our decision has been reached after consultation between the reviewers. Based on these discussions and the individual reviews below, we regret to inform you that your work will not be considered further for publication in *eLife*.

Both reviewers listed a number of strengths of the paper, including the importance of the problem, experimental design and data presentation. However, they were not convinced that the observed frontal modulation of visual responses during speech is actually of retinotopic nature. In addition, there was a concern about the long latencies of responses as well as about the absence of direct evidence that the correlation between activity in frontal and visual regions is a reflection of top-down modulatory influences. The reviews list several other constructive comments and suggestions, which we hope you will find helpful.

*Reviewer #1:*

This is a well-written paper reporting an interesting ECoG investigation of frontal modulation of visual responses during speech perception. The core finding is that visual responses in areas of occipital cortex independently determined to represent the fovea exhibit stronger responses when participants see visual-only-speech stimuli compared to audio-visual speech stimuli. The authors also find that the trial-to-trial amplitude of responses between frontal cortex and visual areas is related for visual areas representing the fovea (functional connectivity), and that the responses in frontal cortex precede (by ~175ms) those in visual cortex.

I have one major concern that would merit collection of some additional data; at present the authors interpretation is not distinguishable from a less interesting alternative. My second concern (2 below) could be addressed through reconsideration of the interpretation of the findings but, would also benefit from some additional data collected from control participants.

1) My principal concern is that the authors have not provided direct evidence that the modulation of visual responses is in fact retinotopic-their argument is that because participant were instructed to fixate the mouth region of the stimuli, and because the effects are observed only in visual areas corresponding to the fovea, the effect is retinotopic. But those data are ambiguous between the authors interpretation and the more bland interpretation that frontal cortex only (ever) modulates visual responses during speech processing in regions of visual cortex corresponding to the fovea. In other words, on the basis of the current set of studies, there is no way to know whether it would be possible for frontal cortex to modulate peripheral representations-knowing that is key for the authors' strong claim that frontal-occipital modulations are in fact retinotopic specific.

Decisive evidence could be brought to bear on this if an additional 1 or 2 patients could be studied using (a) the same exact paradigm to show that they replicate the core finding, and then (b) have participants fixate somewhere besides the mouth. Now it should be the case that the frontal modulation of visual areas is for visual areas that are independently determined to represent the periphery (and specifically, the part of the periphery in which the mouth falls). If the authors obtain those data they would have decisively disambiguated their interpretation from the more prosaic alternative and, would have strong evidence to claim (as they do) that frontal-occipital modulations are indeed retinotopic.

Related to this: note that in Figure 1 and Figure 2, the correlations are largely driven by electrodes that correspond to the visual 'periphery' (i.e., > 5 degrees). If you look at the correlations for only responses in the periphery, there is clear modulation even between 7 degrees and 15 degrees (i.e., in visual regions that definitely are not processing the mouth). Why would there be such modulation according to the authors account? In contrast, on the more prosaic alternative one would expect there to be such a general bias even for nonfoveal visual electrodes.

2) I was surprised at the relatively late latencies in frontal cortex and visual cortex, with the mean frontal latency starting around 400ms and the visual latency around 660ms post stimulus. Given that it takes ~200ms to articulate a syllable, this means that the frontal responses are 2 syllables behind the speech signal, and the visual modulations are 3 syllables behind the speech signal (!). If visual cortex is enhancing its response 3 syllables late, then what possible relevance could that enhancement have on processing? Perhaps a more nuanced discussion is warranted-for instance, could it be that top-down modulation is not useful for improving comprehension of the currently ambiguous syllable but that it will kick in several syllables down the line? If that is the case, it would seem feasible to have psychophysical evidence (from typical subjects) to independently support that. EG listening to normal AV of someone saying 'bi-da-ku-ma-ti-pu…etc', then the audio becomes noisy at syllable n, but the benefit of increased attention to the mouth for disambiguating the speech signal does not kick in until syllable n + 2 in the speech stream.

*Reviewer #2:*

Ozker and colleagues present an experiment in which the contributions of visual and frontal neural populations to speech perception are evaluated. By comparing responses on intracranially implanted electrodes between visual, audio-visual, and auditory speech conditions, they show that visual electrodes show stronger responses to V compared to AV speech. This effect is modulated by whether electrodes have receptive fields that include the location of the mouth in the visual stimulus. Likewise, frontal electrodes show the same pattern, where there are greater responses to V compared to AV speech. In the crucial analyses, they report a correlation between V response amplitudes in frontal and visual electrodes, and a lag between these regions, which they suggest reflects top-down modulation of visual responses by frontal electrodes.

The authors tackle a very interesting and important question for understanding mechanisms of speech perception. I believe the results have important implications both for the audio-visual speech literature, and also for more general theories of top-down modulation effects in speech. The stimuli and experiment are well-designed, and the results are presented in a clear and concise manner. I have the following comments/suggestions:

1) Although I agree with the authors' general premise that frontal regions may provide top-down modulatory signals to other (sensory/perceptual) regions, I am not convinced that they have shown such effects here. First, the claims are based on correlations between response amplitudes across regions during Vis trials. Unfortunately, there is no trial-by-trial behavior to measure comprehension or discrimination effects, which makes it difficult to make strong claims about what variability in response amplitude reflects. They might be able to compensate for this lack of behavior by examining stimulus decoding or classification accuracy. By understanding how well a particular trial can be decoded using only visual information from visual and/or frontal electrodes, it would allow a more direct examination of what frontal electrodes contribute. As it stands, I believe the claims are limited by the classic problem of not knowing whether the correlated activity of two regions is causal, caused by a third region, or correlated due to an unmeasured but otherwise correlated phenomenon.

2) I'm a little confused about the precise claims regarding top-down modulation by frontal neural populations. Is there a reason why the effect should be limited to visual stimuli and visual cortex? Shouldn't there also be a (smaller) effect for auditory stimuli with frontal-auditory electrode pairs? Perhaps there was no coverage of auditory areas in the patients in this cohort, however the authors have published auditory data, so I wonder whether they see analogous effects. This is important because their claims are built on the notion that frontal cortex helps when the stimulus is noisy, which can be true in both modalities.

3) Electrode selection: From the Materials and methods and Results sections, it appears that only one frontal electrode per subject was used. This electrode was then paired with every visual electrode in that subject?

a) The anatomical criterion is not well-motivated. I don't really even understand what the criterion was ("a location near the precentral sulcus, the location of the human frontal eye fields and other attentional control areas" is very vague). How many electrodes per subject were actually implanted in these regions?

b) The functional criterion potentially makes more sense, since it will allow for electrodes with higher SNR to be evaluated. However, in general I don't see why only one electrode per subject was used.

c) Finally, I think that showing negative results for electrodes in other non-frontal regions could help bolster the claims about frontal areas being the specific modulatory signal. As it stands, the vague definition of "frontal" in the paper makes it hard to evaluate what region(s) are even being claimed. Another approach would be to ask whether there is a topography to these very heterogeneous regions, where some frontal regions show stronger modulatory effects than others.

4) In Figure 1, although there is a clear negative correlation, it is also obvious that this correlation is only true for peripheral visual electrodes. If each population is considered separately, there will likely be no effect in the central electrodes. This indicates that within the foveal region, there is no sensitivity to eccentricity, which means that across visual cortex, this is a non-linear effect.a. Related, the correlation for peripheral electrodes is actually quite difficult for me to interpret. First, there are a group of electrodes (around 8-10 deg) that show no difference between V and AV. At even greater eccentricities, the effect actually reverses, where V > AV. First, this suggests that the average responses in Figure 1 obscure some important variability in the nature of the peripheral response. Second, it suggests that the negative correlation with eccentricity in peripheral visual cortex is not an accurate description. Instead, the data suggest that as you go to greater eccentricities, the enhancement effect actually reverses, and that the assumption that the enhancement effect is V > AV is not always true.

5) A similar problem exists in Figure 2. To the extent that there is a negative correlation in central visual electrodes, it is of a different nature than in peripheral electrodes. Whereas the central electrodes may show a correlation where lower eccentricity is associated with a stronger frontal-visual correlation, peripheral electrodes show a reversal with eccentricity, where less eccentric electrodes show a positive correlation, and more eccentric electrodes show a negative correlation. Again, this changes the interpretation of how eccentricity affects these effects. Indeed, the most eccentric peripheral electrodes have about the same predictive power with frontal activity as the least eccentric peripheral electrodes.

[Editors’ note: what now follows is the decision letter after the authors submitted for further consideration.]

Thank you for submitting your article "Retinotopic Modulation of Visual Responses to Speech by Frontal Cortex Measured with Electrocorticography" for consideration by *eLife*. Your article has been reviewed by two peer reviewers, and the evaluation has been overseen by a Reviewing Editor and Andrew King as the Senior Editor. The following individuals involved in review of your submission have agreed to reveal their identity: Bradford Mahon (Reviewer #1); Edward F Chang (Reviewer #2).

The reviewers have discussed the reviews with one another and the Reviewing Editor has drafted this decision to help you prepare a revised submission.

Summary:

The revised manuscript addressed quite well the reservations raised in the previous review. However, there are two remaining issues that both reviewers agreed should be addressed. These are listed below.

Essential revisions:

1) The analysis comparing the frontal-auditory and frontal-visual correlations.

Reviewer 2 states "I strongly suspect that there are electrodes within the cloud shown in Figure 3 that are tuned to the speech sounds on each trial. Are they overall stronger correlations compared to non-relevant auditory electrodes? " Please address this question in your revision.

2) Question about frontal electrodes: Reviewer 2 raises the question about the frontal electrodes, suggesting that they may actually be showing auditory responses. Please address this point in the revision.

*Reviewer #1:*

This revised manuscript addresses directly and decisively the main concern I had from the last round, which was that the core finding was ambiguous between general enhancement of the fovea or specifically the region of visual cortex that processes the mouth. The authors have included a control experiment in which the patient is instructed to fixate on the shoulder of the person in the video--they now show that there is enhancement of visual electrodes processing the mouth even though the mouth is now in the periphery. I consider that lock down evidence for their core claim and am grateful to the authors to taking up that concern and addressing it empirically.

I am also satisfied with the authors' responses to my other concerns, which were relatively minor.

The paper will have a major impact on the field-

*Reviewer #2:*

The authors have addressed my major concerns from the original version of the manuscript. The redone analyses and new data are convincing and help clarify some of the lingering questions about the specificity of the top-down effects. I believe this paper will be an important contribution to the literature.

I have a couple of remaining issues that pertain to the interpretation:

1) I greatly appreciate the authors adding the analysis shown in Figure 3 on effects in auditory electrodes. I think this adds an important dimension to the story. However, I think the analysis comparing the frontal-auditory and frontal-visual correlations is misleading. They claim in the Results section that connectivity was stronger for visual than auditory, however they are actually comparing the TUNED visual electrodes and all auditory electrodes. I strongly suspect that there are electrodes within the cloud shown in Figure 3 that are tuned to the speech sounds on each trial. Are they overall stronger correlations compared to non-relevant auditory electrodes? I think this is important because it addresses the fundamental claim of whether the frontal top-down modulation is modality-specific. If my hypothesis is correct, then combined with the new analysis in response to reviewer 1, I think the story is actually that top-down modulation occurs most strongly for perceptually relevant features (mouth movements and acoustic-phonetic features).

2) I also appreciate the authors clarifying the electrode selection procedures for frontal regions. I am satisfied with SNR as the primary criterion, however I still don't understand the motivation for choosing only 1 electrode per subject. Specifically, I am unsure what these frontal electrodes are actually doing, and I have a suspicion that they are actually showing auditory responses. I think this might be the case given that the average response-locked time-courses (Figure 4) are very similar to auditory responses that have been observed in this dorsal frontal region before (e.g., strong responses above baseline with rapid onsets; see Cheung et al., 2016). In other words, are the electrodes in that region with the highest SNR simply "auditory" electrodes? Would similar effects be observed with the same connectivity analysis between STG and visual electrodes? In my opinion, this is a different interpretation of the effect than if the frontal electrodes were representing some more abstract information about the stimulus or something about attentional modulation, which is the current claim.

---

## [Author Response]

[Editors’ note: the author responses to the first round of peer review follow.]

Both reviewers listed a number of strengths of the paper, including the importance of the problem, experimental design and data presentation. However, they were not convinced that the observed frontal modulation of visual responses during speech is actually of retinotopic nature. In addition, there was a concern about the long latencies of responses as well as about the absence of direct evidence that the correlation between activity in frontal and visual regions is a reflection of top-down modulatory influences. The reviews list several other constructive comments and suggestions, which we hope you will find helpful.

We thank the reviewers for their many helpful criticisms and suggestions. We have completely changed our approach. The major highlights are:

1) We have collected additional ECOG data using a very interesting control experiment suggested by reviewer 1. This experiment allows for the dissociation of the effects of visual field location (central *vs.* peripheral) and the effects of facial location (mouth *vs*. non-mouth) by presenting the talking face stimulus in the visual periphery (Figure 2).

2) As suggested by reviewer 2, we have now measured connectivity between frontal cortex and auditory cortex, using responses measured from 44 auditory cortex electrodes (Figure 3).

3) Both reviewers were confused by our treatment of the eccentricity of each visual electrode as a continuous variable. As they pointed out, it is not clear that there should be a linear relationship between eccentricity in response. These analyses have been removed.

4) Our original focus on visual field eccentricity was confusing. In fact, our main prediction is that visual speech information is concentrated in the mouth region of the talker’s face and the representation of this part of the face should be enhanced. Therefore, we have reanalyzed all of our data using a categorical classification of electrodes as coding either the mouth of the talker’s face or the remainder of the face.

5) Reviewer 2 was confused by our selection criteria for frontal electrodes. We now use a single, simple criterion suggested by the reviewer (signal-to-noise ratio).

6) We have extended our quantitative latency analysis to include all frontal-visual electrode pairs, divided by subjects.

Reviewer #1:

This is a well-written paper reporting an interesting ECoG investigation of frontal modulation of visual responses during speech perception. The core finding is that visual responses in areas of occipital cortex independently determined to represent the fovea exhibit stronger responses when participants see visual-only-speech stimuli compared to audio-visual speech stimuli. The authors also find that the trial-to-trial amplitude of responses between frontal cortex and visual areas is related for visual areas representing the fovea (functional connectivity), and that the responses in frontal cortex precede (by ~175ms) those in visual cortex.I have one major concern that would merit collection of some additional data; at present the authors interpretation is not distinguishable from a less interesting alternative. My second concern (2 below) could be addressed through reconsideration of the interpretation of the findings but, would also benefit from some additional data collected from control participants.1) My principal concern is that the authors have not provided direct evidence that the modulation of visual responses is in fact retinotopic-their argument is that because participant were instructed to fixate the mouth region of the stimuli, and because the effects are observed only in visual areas corresponding to the fovea, the effect is retinotopic. But those data are ambiguous between the authors interpretation and the more bland interpretation that frontal cortex only (ever) modulates visual responses during speech processing in regions of visual cortex corresponding to the fovea. In other words, on the basis of the current set of studies, there is no way to know whether it would be possible for frontal cortex to modulate peripheral representations-knowing that is key for the authors' strong claim that frontal-occipital modulations are in fact retinotopic specific.Decisive evidence could be brought to bear on this if an additional 1 or 2 patients could be studied using (a) the same exact paradigm to show that they replicate the core finding, and then (b) have participants fixate somewhere besides the mouth. Now it should be the case that the frontal modulation of visual areas is for visual areas that are independently determined to represent the periphery (and specifically, the part of the periphery in which the mouth falls). If the authors obtain those data they would have decisively disambiguated their interpretation from the more prosaic alternative and, would have strong evidence to claim (as they do) that frontal-occipital modulations are indeed retinotopic.

We agree with the reviewer that our initial study confounded the location of enhancement: since the talker’s mouth was always foveal, the observed enhancement could be specific to the location of the talker’s mouth, or specific to the center of gaze. We thank the reviewer for suggesting a very interesting control experiment. We programmed the experiment and have collected data using this design (Figure 2). The results show that enhancement is specific to the mouth of the talker, not the center of gaze.

Related to this: note that in Figure 1 and Figure 2, the correlations are largely driven by electrodes that correspond to the visual 'periphery' (i.e., > 5 degrees). If you look at the correlations for only responses in the periphery, there is clear modulation even between 7 degrees and 15 degrees (i.e., in visual regions that definitely are not processing the mouth). Why would there be such modulation according to the authors account? In contrast, on the more prosaic alternative one would expect there to be such a general bias even for nonfoveal visual electrodes.

We agree and this analysis has been removed from the paper; see points 3 and 4, above.

2) I was surprised at the relatively late latencies in frontal cortex and visual cortex, with the mean frontal latency starting around 400ms and the visual latency around 660ms post stimulus. Given that it takes ~200ms to articulate a syllable, this means that the frontal responses are 2 syllables behind the speech signal, and the visual modulations are 3 syllables behind the speech signal (!). If visual cortex is enhancing its response 3 syllables late, then what possible relevance could that enhancement have on processing? Perhaps a more nuanced discussion is warranted-for instance, could it be that top-down modulation is not useful for improving comprehension of the currently ambiguous syllable but that it will kick in several syllables down the line? If that is the case, it would seem feasible to have psychophysical evidence (from typical subjects) to independently support that. EG listening to normal AV of someone saying 'bi-da-ku-ma-ti-pu…etc', then the audio becomes noisy at syllable n, but the benefit of increased attention to the mouth for disambiguating the speech signal does not kick in until syllable n + 2 in the speech stream.

We apologize that the latency values reported in the original manuscript were confusing. The times reported in the original manuscript were calculated from the beginning of the video clip rather than the beginning of speech. We now report latencies with respect to this. For instance, “an average latency difference of 230 ms (frontal *vs.* visual: 257 ± 150 ms *vs.* 487 ± 110 ms *vs.* with respect to auditory speech onset, t_34_ = 4, p = 10^-4^).”

While attentional enhancement may take some time to “kick in”, it can provide a sustained boost in processing efficiency. While in our experiments, a single word was presented, leading to a brief attentional response in frontal cortex, in a longer speech stream, attention to the mouth would be long-lasting, producing a sustained response in frontal cortex and sustained enhancement of visual speech information in visual cortex.

Reviewer #2:

Ozker and colleagues present an experiment in which the contributions of visual and frontal neural populations to speech perception are evaluated. By comparing responses on intracranially implanted electrodes between visual, audio-visual, and auditory speech conditions, they show that visual electrodes show stronger responses to V compared to AV speech. This effect is modulated by whether electrodes have receptive fields that include the location of the mouth in the visual stimulus. Likewise, frontal electrodes show the same pattern, where there are greater responses to V compared to AV speech. In the crucial analyses, they report a correlation between V response amplitudes in frontal and visual electrodes, and a lag between these regions, which they suggest reflects top-down modulation of visual responses by frontal electrodes.The authors tackle a very interesting and important question for understanding mechanisms of speech perception. I believe the results have important implications both for the audio-visual speech literature, and also for more general theories of top-down modulation effects in speech. The stimuli and experiment are well-designed, and the results are presented in a clear and concise manner. I have the following comments/suggestions:1) Although I agree with the authors' general premise that frontal regions may provide top-down modulatory signals to other (sensory/perceptual) regions, I am not convinced that they have shown such effects here. First, the claims are based on correlations between response amplitudes across regions during Vis trials. Unfortunately, there is no trial-by-trial behavior to measure comprehension or discrimination effects, which makes it difficult to make strong claims about what variability in response amplitude reflects. They might be able to compensate for this lack of behavior by examining stimulus decoding or classification accuracy. By understanding how well a particular trial can be decoded using only visual information from visual and/or frontal electrodes, it would allow a more direct examination of what frontal electrodes contribute. As it stands, I believe the claims are limited by the classic problem of not knowing whether the correlated activity of two regions is causal, caused by a third region, or correlated due to an unmeasured but otherwise correlated phenomenon.

We thank the reviewer for this suggestion. We attempted a decoding analysis using a number of methods with our data. Unfortunately, with only 8 to 16 trials for each of the presented words, there was not enough data to perform a reliable classification analysis.

2) I'm a little confused about the precise claims regarding top-down modulation by frontal neural populations. Is there a reason why the effect should be limited to visual stimuli and visual cortex? Shouldn't there also be a (smaller) effect for auditory stimuli with frontal-auditory electrode pairs? Perhaps there was no coverage of auditory areas in the patients in this cohort, however the authors have published auditory data, so I wonder whether they see analogous effects. This is important because their claims are built on the notion that frontal cortex helps when the stimulus is noisy, which can be true in both modalities.

We thank the reviewer for this suggestion, we agree that an examination of frontal-auditory connectivity is interesting and important. We have performed this analysis and now report it in Figure 3.

3) Electrode selection: From the Materials and methods and Results sections, it appears that only one frontal electrode per subject was used. This electrode was then paired with every visual electrode in that subject?a) The anatomical criterion is not well-motivated. I don't really even understand what the criterion was ("a location near the precentral sulcus, the location of the human frontal eye fields and other attentional control areas" is very vague). How many electrodes per subject were actually implanted in these regions?b) The functional criterion potentially makes more sense, since it will allow for electrodes with higher SNR to be evaluated. However, in general I don't see why only one electrode per subject was used.c) Finally, I think that showing negative results for electrodes in other non-frontal regions could help bolster the claims about frontal areas being the specific modulatory signal. As it stands, the vague definition of "frontal" in the paper makes it hard to evaluate what region(s) are even being claimed. Another approach would be to ask whether there is a topography to these very heterogeneous regions, where some frontal regions show stronger modulatory effects than others.

We thank the reviewer for these helpful suggestions.

a) We eliminated the anatomical criterion.

b) As suggested by the reviewer, we now use the SNR as our sole criterion, picking the frontal electrode with the highest SNR in each subject. We also used the two electrodes with the highest SNR in each subject and found similar results. In general, there is a great deal of heterogeneity in frontal electrodes and examining only electrodes with the most robust response helps ensure reliability.

Subsection “Frontal Cortex Electrode Selection”: We recorded from 179 electrodes located in lateral frontal cortex, defined as the lateral convexity of the hemisphere anterior to the central sulcus. 44 out of 179 electrodes showed a significant (q < 0.01, false-discovery rate corrected) broadband response to *AV* speech. In each of the 8 subjects, we selected the single frontal electrode that showed the largest broadband response to *AV* speech, measured as the signal-to-noise ratio (μ/σ).

c) We now show that connectivity between frontal cortex and mouth electrodes is significant greater than connectivity between frontal cortex and non-mouth electrodes or connectivity between frontal cortex and auditory cortex.

4) In Figure 1, although there is a clear negative correlation, it is also obvious that this correlation is only true for peripheral visual electrodes. If each population is considered separately, there will likely be no effect in the central electrodes. This indicates that within the foveal region, there is no sensitivity to eccentricity, which means that across visual cortex, this is a non-linear effect.a. Related, the correlation for peripheral electrodes is actually quite difficult for me to interpret. First, there are a group of electrodes (around 8-10 deg) that show no difference between V and AV. At even greater eccentricities, the effect actually reverses, where V > AV. First, this suggests that the average responses in Figure 1 obscure some important variability in the nature of the peripheral response. Second, it suggests that the negative correlation with eccentricity in peripheral visual cortex is not an accurate description. Instead, the data suggest that as you go to greater eccentricities, the enhancement effect actually reverses, and that the assumption that the enhancement effect is V > AV is not always true.5) A similar problem exists in Figure 2. To the extent that there is a negative correlation in central visual electrodes, it is of a different nature than in peripheral electrodes. Whereas the central electrodes may show a correlation where lower eccentricity is associated with a stronger frontal-visual correlation, peripheral electrodes show a reversal with eccentricity, where less eccentric electrodes show a positive correlation, and more eccentric electrodes show a negative correlation. Again, this changes the interpretation of how eccentricity affects these effects. Indeed, the most eccentric peripheral electrodes have about the same predictive power with frontal activity as the least eccentric peripheral electrodes.

We agree completely and this analysis has been removed from the paper; see points 3 and 4 in the first section of the response.

[Editors' note: the author responses to the re-review follow.]

Essential revisions:1) The analysis comparing the frontal-auditory and frontal-visual correlations.Reviewer 2 states "I strongly suspect that there are electrodes within the cloud shown in Figure 3 that are tuned to the speech sounds on each trial. Are they overall stronger correlations compared to non-relevant auditory electrodes? " Please address this question in your revision.

While the primary focus of our manuscript is on frontal modulation of visual cortex, we agree that auditory cortex provides an important comparison. We have performed new analyses and created a new figure (Figure 5) to provide more information. We now report in the Results section:

“While our primary focus was on frontal modulation of visual cortex, auditory cortex was analyzed for comparison. 44 electrodes located on the superior temporal gyrus responded to *AV* speech (Figure 5). As expected, these electrodes showed little response to *Vis* speech but a strong response to *Aud* speech, with a maximal response amplitude of 148percent at 67 ms after auditory speech onset. While visual cortex showed a greater response to *Vis* speech than to *AV* speech, there was no significant difference in the response of auditory cortex to *Aud* and *AV* speech, as confirmed by the LME model (p = 0.8, Supplementary Table 4). Next, we analyzed the connectivity between frontal cortex and auditory electrodes. The LME revealed a significant effect of stimulus, driven by weaker connectivity in the *Vis* condition in which no auditory speech was presented (p = 0.02; Supplementary Table 5). Overall, the connectivity between frontal cortex and auditory cortex was weaker than the connectivity between frontal cortex and visual cortex (0.04 for frontal-auditory *vs.* 0.23 for frontal-visual mouth electrodes; p = 0.001, unpaired t-test).

In visual cortex, a subset of electrodes (those representing the mouth) showed a greater response in the *Vis-AV* contrast and greater connectivity with frontal cortex. To determine if the same was true in auditory cortex, we selected the STG electrodes with the strongest response in the *Aud-AV* contrast (Figure 5). Unlike in visual cortex, in which there was anatomical localization of mouth-representing electrodes on the occipital pole, auditory electrodes with the strongest response in the *Aud-AV* contrast did not show clear anatomical clustering (Figure 5). Electrodes with the electrodes with the strongest response in the *Aud-AV* contrast had equivalent connectivity with frontal cortex as other STG electrodes (ρ = -0.04 *vs. roh* = 0.06, unpaired t-test = 0.9, p = 0.3). This was the opposite of the pattern observed in visual cortex. In order to ensure that signal amplitude was not the main determinant of connectivity, we selected the STG electrodes with the highest signal-to-noise ratio in the AV condition. These electrodes had equivalent connectivity with frontal cortex as other STG electrodes (ρ = -0.002 *vs.* ρ =0.05, unpaired t-test = 0.4, p = 0.7).”

2) Question about frontal electrodes: Reviewer 2 raises the question about the frontal electrodes, suggesting that they may actually be showing auditory responses. Please address this point in the revision.

We now write in the Discussion section:

“The analysis of auditory cortex responses provides an illuminating contrast with the visual cortex results. While removing auditory speech increased visual cortex response amplitudes and frontal-visual connectivity, removing visual speech did not change auditory cortex response amplitudes or connectivity. A simple explanation for this is that auditory speech is easily intelligible without visual speech, so that no attentional modulation is required. In contrast, perceiving visual-only speech requires speechreading, a difficult task that demands attentional engagement. An interesting test of this idea would be to present auditory speech with added noise. Under these circumstances, attentional engagement would be expected in order to enhance the intelligibility of the noisy auditory speech, with a neural manifestation of increased response amplitudes in auditory cortex and increased connectivity with frontal cortex. The interaction of stimulus and task also provides an explanation for the frontal activations in response to the speech stimuli. The human frontal eye fields show rapid sensory-driven responses, with latencies as short as 24 ms to auditory stimuli and 45 ms to visual stimuli (Kirchner et al., 2009). Following initial sensory activation, task demands modulate frontal function, with demanding tasks such as processing visual-only or noisy auditory speech or resulting in enhanced activity (Cheung et al., 2016, Cope et al., 2017).”